# Anti-SARS-CoV-2 antibody kinetics up to 6 months of follow-up: Result from a nation-wide population-based, age stratified sero-epidemiological prospective cohort study in India

Puneet Misra[1☉*], Arvind Kumar Singh[2‡], Baijayantimala Mishra[3‡], Bijayini Behera[2‡], Binod Kumar Patro[2‡], Guruprasad R. Medigeshi[4‡], Hari Shanker Joshi[5‡], Mohammad Ahmad[6], Pradeep Kumar Chaturvedi[7‡], Palanivel Chinnakali[8‡], Partha Haldar[1‡], Mohan Bairwa[1‡], Pradeep Kharya[5‡], Rahul Dhodapkar[9‡], Ramashankar Rath[5], Randeep Guleria[10], Sanjay Kumar Rai[1], Sitanshu Sekhar Kar[8‡], Shashi Kant[1☉], Sonali Sarkar[8‡], Subrata Baidya[11‡], Suneeta Meena[12‡], Suprakash Mandal[1], Surekha Kishore[13], Tapan Majumder[14‡], Vivek Hada[15‡]

1 Centre for Community Medicine, All India Institute of Medical Sciences, New Delhi, India, 2 Department of Community and Family Medicine, All India Institute of Medical Sciences, Bhubaneshwar, Odisha, India, 3 Department Microbiology, All India Institute of Medical Sciences, Bhubaneshwar, Odisha, India, 4 Translational Health Science and Technology Institute, Faridabad, India, 5 Department of Community and Family Medicine, All India Institute of Medical Sciences, Gorakhpur, India, 6 World Health Organization, WHO Country Office, New Delhi, India, 7 Department of Reproductive Biology, All India Institute of Medical Sciences, New Delhi, India, 8 Department of Community Medicine, Jawaharlal Institute of Postgraduate Medical Education and Research, Puducherry, India, 9 Department of Microbiology, Jawaharlal Institute of Postgraduate Medical Education and Research, Puducherry, India, 10 All India Institute of Medical Sciences, New Delhi, India, 11 Department of Community Medicine, Agartala Government Medical College, Agartala, India, 12 Department of Laboratory Medicine, All India Institute of Medical Sciences, New Delhi, India, 13 All India Institute of Medical Sciences, Gorakhpur, India, 14 Department of Microbiology, Agartala Government Medical College, Agartala, India, 15 Department of Microbiology, All India Institute of Medical Sciences, Gorakhpur, Uttar Pradesh, India

☉ These authors contributed equally to this work.
‡ AKS, BM, BB, BKP, GRM, HSJ, PKC, PC, PH, MB, PK, RD, SSK, SS, SB, SM, TM and VH also contributed equally to this work.
* doctormisra@gmail.com

**Data Availability Statement:** The data set has identifiable information of family and patients along

## Abstract

Repeated serological testing tells about the change in the overall infection in a community. This study aimed to evaluate changes in antibody prevalence and kinetics in a closed cohort over six months in different sub-populations in India. The study included 10,000 participants from rural and urban areas in five states and measured SARS-CoV-2 antibodies in serum in three follow-up rounds. The overall seroprevalence increased from 73.9% in round one to 90.7% in round two and 92.9% in round three. Among seropositive rural participants in round one, 98.2% remained positive in round two, and this percentage remained stable in urban and tribal areas in round three. The results showed high antibody prevalence that increased over time and was not different based on area, age group, or sex. Vaccinated individuals had higher antibody prevalence, and nearly all participants had antibody positivity for up to six months.

with lot of medical history which are potentially sensitive. The restriction has been imposed by: Institute Ethics Committee Therefore, the data may be requested with the corresponding author Dr Puneet Misra : doctormisra@gmail.com or with the Dean (Research) of the investigating institute to the following address: Non-author point of contact: Dr. Jeewan Singh Titiyal Dean, Research Research Section All India Institute of Medical Sciences, Ansari Nagar East New Delhi-110029 Mail: researchsectionaiims@gmail.com.

**Funding:** This work was supported by a research grant (Ref No: 2020/1085497, Purchase Order: 202630166) from the World Health Organisation (WHO) Country Office, New Delhi 110016, India. The funders had no role in study design, data collection and analysis, decision to publish, or preparation of the manuscript.

**Competing interests:** The authors have declared that no competing interests exist.

**Abbreviations:** AIIMS, All India Institute of Medical Sciences; COVID, Coronavirus Disease; ELISA, Enzyme-linked Immunosorbent Assay; NCR, Delhi National Capital Region; SARS-CoV-2, Severe Acute Respiratory Syndrome Coronavirus-2; WHO, World Health Organization.

## Introduction

The COVID-19 pandemic remains an important public health issue since 2019. Since its onset, several waves were seen in different countries throughout the world in different time periods [1, 2]. In India, the first wave was seen from March to November 2020. The second wave in the form of the delta variant, from March to June 2021, led to a massive surge of symptomatic COVID, hospitalization, and fatality [3]. There was a subsequent third wave of less virulent omicron variant at the end of 2021. For almost 3 years India experienced this series of the pandemic waves, which had impact on population health. India started the COVID-19 vaccination in January 2021; initially for the health care workers, and frontline workers followed by the elderly population, and finally for all adults [4].

India, as well as most parts of the world, has entered the phase of endemicity where a substantial proportion of the population is positive for SARS-CoV-2 antibody [5]. Varying levels of seropositivity were found among the population of different states of India. The seropositivity status of a population indicates the extent of past infection, the indirect protection from subsequent infection, the proportion of the susceptible population, etc. Similarly, repeated serological testing tells about the change in the overall infection in a community. There were very few studies in cohort design assessing the antibody kinetics for a longer duration. Therefore, we aimed to assess the change in antibody prevalence and its kinetics in a closed cohort over six months across the country in different sub-population in India.

## Methodology

### Study design

This was a population-based, multi-centric, age-stratified prospective cohort study under WHO (World Health Organisation) Unity protocols for the SARS-CoV-2 sero-surveillance [6]. A total of three rounds of follow-up were done for the same cohort of the study participants viz. at baseline, 3rd months, and 6th months.

### Study setting and patient selection

The study was conducted at five selected study sites in India. The study sites were Delhi, Bhubaneswar, Gorakhpur, Pondicherry, and Agartala situated in the state of Delhi, Odisha, Uttar Pradesh, Pondicherry, and Tripura respectively.

The study population were from both the rural and urban areas of each site except at the Agartala site where the tribal population was included in place of the urban population. For Delhi site, an urban resettlement colony consisting mostly of a population from low socio-economic strata in the south Delhi district was chosen to represent urban population. The rural population was selected from Ballabgarh block in the Faridabad district of nearby Haryana state. Faridabad district had contiguous border with Delhi state and came under Delhi National Capital Region (Delhi NCR). Ballabgarh block was the rural field practice area of the investigating site spread across 50 square kilometres with 28 villages with a population of 102,000 as per 2021 data. The Bhubaneswar site in south-Eastern India included an urban area under the Bhubaneswar Municipal Corporation (BMC) and a rural area at selected villages of adjacent Khordha districts at a distance of 25–40 kilometres from the site institute. Gorakhpur was a city situated in Uttar Pradesh, a state in Northern India. The city was a major transit point of surface transport near the India-Nepal international border. The rural area was in the vicinity of the city with an average distance of 16–72 kilometres from the city centre whereas the urban area was in the centre of the city in a selected municipality block. In north-eastern India, the Agartala site was situated in the state of Tripura. This site included the rural and

tribal population from the selected villages situated at a distance of 16–30 km from the city centre. In southern India, the Puducherry site was a Union Territory of an area of 20 square kilometres where selected municipality wards and villages were included for the urban and rural population.

## Sample size and sampling strategy

In each of the study sites, population both from an urban and rural area (except Agartala: rural and tribal) were included. Individual villages in the rural area and municipality wards in the urban area were considered as a cluster. From each rural and urban area, 25 clusters were selected purposively. From each cluster, 40 participants were selected to finally achieve 1,000. Therefore, each study site had a sample size of 2,000 and the total sample size for the study was 10,000. The recruitment of the participants started from a meeting point of multiple lanes of a cluster preferably at the centre. The rotating pencil method was used to identify the first starting lane where $\geq 10$ consecutive families were approached. From those houses, at least 40 participants having age more than equal to 1 year were recruited. The rule of the left was adopted to move further at the end of any lane. The same participants were followed up to collect biological samples at baseline, 3rd month and 6th month.

## Outcome measures

Our main outcome measure was the presence or absence of antibodies against the SARS-CoV-2 virus in serum. It was assessed by standard Enzyme-linked Immunoassay (ELISA) (Kit: WANTAI SARS-CoV-2 Ab ELISA kit, Wantai SARS-CoV-2 Diagnostics) as per the manufacturer's protocol. The kit captured human serum total antibody (IgM + IgG) against the receptor binding domain of spike protein of the SARS-Cov-2 virus and detected the antibody qualitatively. The sensitivity and specificity of the test kit were 94.4% and 100% respectively [7]. Those serum specimens having a ratio of observed absorbance to cut-off (O/C) $\geq 1.0$ was taken as reactive for the antibody.

## Biological specimen collection and handling

Three to five millilitre of venous blood sample was collected by aseptic venepuncture from each participant. The blood sample was centrifuged (at 3000 rpm for 10 minutes) within two hours of collection. Serum was stored at 2–8 degree Celsius for laboratory analysis within seven days, otherwise at -80 degrees Celsius for long-term storage. The standard operating procedure for handling biological samples was followed.

## Other variable

We interviewed the adult participants and the guardian of the minor participants to obtain basic socio-demographic details like age, sex, residence, educational status, occupation, any substance use, any clinical symptoms experienced in the past three months, history of SARS-CoV-2 laboratory test, COVID-19 vaccination status, etc.

## Data collection tool and data quality management

We used electronic tablet-based Epi-Collect 5 data collection software to enter the data of participants' interviews as well as data of laboratory results. The web portal of Epi-Collect 5 was used to do real-time based monitoring of the progress, identifying any data entry error, incompleteness, and data mismatch simultaneously during the period of data collection. The appropriate prompt action was taken to resolve issues that appeared at any of the study site. Periodic

data download and cross-checking were also done to find any discrepancies by a designated data manager. A weekly progress report was obtained and discussed with all the study sites to ensure timely data collection and maintaining quality. Apart from this periodic refresher training and study site review meetings were conducted to address the site-specific issues and ensure timely dissemination of quality data. Standard state-specific COVID-appropriate guideline was followed during the data collection. The uploaded data were exported to Microsoft Excel format and merged with the subsequent round of data with the help of unique identification numbers.

### Data analysis

Data data analysis was done by STATA Version 12 (STATA Corporation, Texas, USA) statistical software. Data cleaning was done with the help of both Microsoft Excel and STATA by a qualified data manager as well as by the study investigators. Descriptive statistical analysis was done and the result was expressed by frequency and proportion for categorical variables and mean (SD), 95% confidence interval (CI) for the continuous variable. The seroprevalence was presented by percentage and with 95% CI by on the study site, round, urban-rural area, age group, sex, according to symptoms, and vaccination status.

### Ethical approval and consent to participate

Ethical approval was obtained from all five investigating institutes (Letter No. For AIIMS, New Delhi: **IEC-959/04.09.2020,** AIIMS Bhubaneswar: **T/EMF/CM&FM/20/44**, JIPMER Puducherry: **JIP/IEC/2020/248**, AIIMS Gorakhpur: **IHEC/AIIMS-GKP/BMR/01/22**, Agartala: **F.4 (5–234)/AGMC/ACADEMIC/IEC MEETING**). We obtained informed written consent, assent, and consent from the parents/guardians of the participants who were under the legal age for giving consent.

## Results

The data collection period was from March 2021 to August 2021 in round one, from May 2021 to December 2021 in round two, and from August 2021 to January 2022 in round three. Since multiple centre were at different phase of the data collection due to their local operational pattern, the data collection period in different rounds were overlapping. (S1 Table in S1 File).

The total number of participants in round one was 10,110 for all sites clubbed together. In the subsequent rounds, 6,503 (64.3%) remained in round two and 5,564 (55.0%) in round three. The highest proportion of participants who remained in the cohort till round three was at the Bhubaneswar site (73.6%), whereas the minimum was at the Pondicherry site (33.1%). (S2 Table in S1 File).

In round two higher proportion of participants from the rural area remained in the study compared to the urban participants across all sites. In contrast to this, higher attrition rate was seen in the rural area in round three across all sites. (S2 Table in S1 File). The overall proportion of males in all three rounds was lower than females. The recruited participants were mostly older than 10 years and younger than 60 years. This age group formed nearly 70% of all the participants and remained stable across the three rounds. (S3 Table in S1 File).

Of those lost to follow-up in urban areas, 655 (50%) participants in round two and 469 (32.3%) participants in round three were not traceable even after three domiciliary visits. The second most common reason for attrition was refusal to continue participation in the study (37.2% in urban, 47.3% in rural, and 98.9% in tribal areas). In round three, the proportion who refused to participate was higher than those who couldn't be traced. (S4 Table in S1 File).

The overall seroprevalence was 73.9% (95% CI: 73.1–74.8) in round one which increased to 90.7% (95% CI: 89.9–91.4) in round two and 92.9% (95% CI: 92.2–93.6) in round three. The highest seroprevalence in round one was at the Gorakhpur site (91.7%, 95% CI: 90.5–92.9) whereas the lowest was at the Bhubaneswar site (64.1%, 95% CI: 62.0–66.3) giving a wide range of seroprevalence across the study sites. The seroprevalence in round two was similar across all the sites ranging from 83.8% (95% CI: 82.0–85.6) to 94.9% (95% CI: 93.5–96.1). The range of seroprevalence further narrowed down in round three. (Table 1).

In round one the seroprevalence was higher in the urban area across all the sites; the overall seroprevalence in urban area was 81.5% and rural 69.6% and 66.7% in the tribal area. In round two these were 91.1% in urban, 90.4% in rural, and 90.4% in tribal area. Whereas, in round three it was 91.8%, 94.1%, and 91.6% in urban, rural, and tribal area respectively. (S5 Table in S1 File).

Sex-wise seroprevalence in round one was 73.8% among males, and 74.0% among females. In round two it was 89.4% among males, and 91.7% among females. In round three, the seroprevalence among males was 91% and among females, it was 94.3%. (Table 2).

The seroprevalence among participants aged less than 18 years was 67.1%, 82.3%, and 85.1% in rounds one, two, and three, respectively. The seroprevalence among the participants aged 18 years or older was 75.2%, 92.2%, and 94.2% in the three rounds, respectively. (Table 3).

The proportion of symptomatic individuals among the seropositive was 26.5% overall in round one. The proportion of symptomatic declined slightly in round two (25.1%) and further declined to 20.1% in round three. (S6 Table in S1 File).

The proportion of vaccinated individuals in round one was 31.5% which increased to 58.8% in round two and 61.0% in round three. The seroprevalence in round one among the vaccinated individuals was 86.5% whereas among unvaccinated it was 68.1%. In subsequent rounds, the prevalence reached nearly 95% among vaccinated and around 85% among unvaccinated. (Table 4).

Among the seropositive rural participants in round one, 98.2% remained positive in round 2. This percentage remained stable in the urban and tribal areas and in round three also. Among the seronegative participants, 73.5% in rural, 60.5% in the urban area, and 78.2% in

**Table 1. Distribution of SARS-CoV-2 seropositive participants by site and round.**

| Study site | Round One n (%) | | Round Two n (%) | | Round Three n (%) | |
|---|---|---|---|---|---|---|
| | Sample size | Seropositive n (%) 95% CI | Sample size | Seropositive n (%) 95% CI | Sample size | Seropositive n (%) 95% CI |
| **Delhi** | 2060 | 1373 (66.6) (64.6–68.7) | 1569 | 1485 (94.7) (93.4–95.7) | 1395 | 1336 (95.8) (94.5–96.7) |
| **Bhubaneswar** | 2000 | 1283 (64.1) (62.0–66.3) | 1704 | 1429 (83.8) (82.0–85.6) | 1473 | 1284 (87.2) (85.3–88.8) |
| **Gorakhpur** | 2010 | 1845 (91.7) (90.5–92.9) | 1151 | 1093 (94.9) (93.5–96.1) | 937 | 903 (96.4) (94.9–97.5) |
| **Agartala** | 2000 | 1254 (62.7) (60.5–64.8) | 856 | 776 (90.7) (88.5–92.5) | 1095 | 1020 (93.2) (91.5–94.6) |
| **Pondicherry** | 2040 | 1719 (84.2) (82.6–85.8) | 1222 | 1114 (91.6) (89.4–92.6) | 664 | 627 (94.4) (92.4–96.1) |
| **Total** | 10110 | 7474 (73.9) (73.1–74.8) | 6503 | 5897 (90.7) (89.9–91.4) | 5564 | 5170 (92.9) (92.2–93.6) |

**Table 2. Distribution of SARS-CoV-2 seropositive participants by site, round, and sex.**

| | | Agartala | | Bhubaneshwar | | Delhi | | Gorakhpur | | Puducherry | | Total | |
|---|---|---|---|---|---|---|---|---|---|---|---|---|---|
| | | Sample size | Sero-positive | Sample size | Sero-positive | Sample size | Sero-positive | Sample size | Sero-positive | Sample size | Sero-positive | Sample size | Sero-positive |
| | | n | n | n | n | n | n | n | n | n | n | N | n |
| | | | (%) | | (%) | | (%) | | (%) | | (%) | | (%) |
| **Male** | Round one | 803 | 508 | 926 | 596 | 930 | 588 | 1035 | 955 | 834 | 697 | 4528 | 3343 |
| | | | (63.3) | | (64.4) | | (63.2) | | (92.2) | | (83.6) | | (73.8) |
| | Round Two | 300 | 268 | 780 | 642 | 682 | 631 | 558 | 530 | 447 | 402 | 2767 | 2473 |
| | | | (89.3) | | (82.3) | | (92.5) | | (95.0) | | (89.9) | | (89.4) |
| | Round Three | 397 | 365 | 656 | 553 | 573 | 538 | 423 | 406 | 226 | 208 | 2275 | 2070 |
| | | | (91.9) | | (84.3) | | (93.9) | | (96.0) | | (92.0) | | (91.0) |
| **Female** | Round one | 1197 | 746 | 1074 | 687 | 1130 | 785 | 975 | 890 | 1206 | 1022 | 5582 | 4131 |
| | | | (62.3) | | (63.9) | | (69.5) | | (91.4) | | (84.7) | | (74.0) |
| | Round Two | 556 | 508 | 924 | 786 | 887 | 854 | 593 | 563 | 775 | 712 | 3736 | 3424 |
| | | | (91.4) | | (85.1) | | (96.3) | | (94.9) | | (91.9) | | (91.7) |
| | Round Three | 697 | 655 | 817 | 731 | 822 | 798 | 513 | 497 | 440 | 422 | 3289 | 3101 |
| | | | (93.8) | | (89.5) | | (97.1) | | (96.7) | | (95.7) | | (94.3) |

the tribal area were converted to seropositive in round two. In round three, among seropositive, 45.6% in rural, 25.7% in urban, and 33.3% in tribal area seroconverted to positive. (Table 5).

## Discussion

This nationwide multicentric population-based seroepidemiological cohort study attempted to find the serum antibody prevalence against SARS-CoV-2 virus among the general population up to 6[th] month after the initial serological assessment. The data collection period in round one coincided with the before and after the second wave of SARS-CoV-2 infection in India.

**Table 3. Distribution of SARS-CoV-2 seropositive of participants by site, round and age group (<18 years and ≥18 years).**

| Age category | | Agartala | | Bhubaneshwar | | Delhi | | Gorakhpur | | Puducherry | | Total | |
|---|---|---|---|---|---|---|---|---|---|---|---|---|---|---|
| | | Sample size | Sero-positive | Sample size | Sero-positive | Sample size | Sero-positive | Sample size | Sero-positive | Sample size | Sero-positive | Sample size | Sero-positive |
| | | n | n | n | n | n | n | n | n | n | n | N | n |
| | | | (%) | | (%) | | (%) | | (%) | | (%) | | (%) |
| **< 18 years** | Round one | 230 | 108 | 320 | 175 | 281 | 183 | 339 | 282 | 372 | 286 | 1542 | 1034 |
| | | | (47.0) | | (54.7) | | (65.1) | | (83.2) | | (76.9) | | (67.1) |
| | Round Two | 75 | 58 | 283 | 218 | 223 | 203 | 193 | 164 | 218 | 173 | 992 | 816 |
| | | | (77.3) | | (77.0) | | (91.0) | | (85.4) | | (79.4) | | (82.3) |
| | Round Three | 111 | 95 | 244 | 198 | 183 | 164 | 165 | 148 | 92 | 73 | 795 | 678 |
| | | | (84.8) | | (81.2) | | (89.1) | | (89.7) | | (79.4) | | (85.1) |
| **≥ 18 years** | Round one | 1770 | 1146 | 1680 | 1108 | 1779 | 1190 | 1671 | 1563 | 1668 | 1433 | 8568 | 6440 |
| | | | (64.8) | | (66.0) | | (66.9) | | (93.5) | | (86.0) | | (75.2) |
| | Round Two | 781 | 718 | 1421 | 1210 | 1346 | 1282 | 959 | 929 | 1004 | 941 | 5511 | 5080 |
| | | | (91.9) | | (85.2) | | (95.3) | | (96.9) | | (93.7) | | (92.2) |
| | Round Three | 983 | 925 | 1229 | 1086 | 1210 | 1172 | 772 | 755 | 575 | 557 | 4769 | 4495 |
| | | | (94.1) | | (88.4) | | (96.8) | | (97.8) | | (96.9) | | (94.2) |

**Table 4. Distribution of SARS-CoV-2 seropositive of participants by round, vaccination status and site.**

| Round | Vaccination status | Agartala | | Bhubaneshwar | | Delhi | | Gorakhpur | | Puducherry | | Total | |
|---|---|---|---|---|---|---|---|---|---|---|---|---|---|
| | | Sample | Sero-positive n (%) | Sample | Sero-positive n (%) | Sample | Sero-positive n (%) | Sample | Sero-positive n (%) | Sample | Sero-positive n (%) | Sample* | Sero-positive n (%) |
| Round one | Yes | 728 | 621 (85.3) | 778 | 582 (74.8) | 214 | 148 (69.2) | 727 | 706 (97.1) | 738 | 699 (94.7) | 3186 | 2756 (86.5) |
| | No | 1272 | 633 (49.8) | 1222 | 701 (57.4) | 1844 | 1224 (66.4) | 1275 | 1132 (88.8) | 1301 | 1019 (78.3) | 6915 | 4709 (68.1) |
| Round Two | Yes | 601 | 570 (94.8) | 1072 | 928 (86.6) | 635 | 614 (96.7) | 745 | 729 (97.9) | 769 | 756 (98.3) | 3822 | 3597 (94.1) |
| | No | 255 | 206 (80.8) | 631 | 499 (79.1) | 933 | 871 (93.3) | 401 | 359 (89.5) | 452 | 358 (79.0) | 2672 | 2293 85.8) |
| Round Three | Yes | 763 | 740 (97.0) | 1144 | 1013 (88.5) | 876 | 856 (97.7) | 712 | 696 (97.7) | 505 | 496 (98.2) | 4000 | 3801 (95.0) |
| | No | 332 | 280 (84.3) | 328 | 270 (82.3) | 510 | 472 (92.6) | 223 | 205 (91.9) | 162 | 134 (82.7) | 1555 | 1361 (87.5) |

*9 participants didn't had idea of their vaccination status across all three rounds

The round two data were collected during and after the second wave but before the third wave whereas the round three data collection was done before and after the third wave. At the end of three round more than 50% participants remained in the study. The rural area participants remained more in the first follow-up due which may be due to the inherent non-migratory nature and integrity of the area but during the third round lost to follow-up was more compared to the urban area. This may be due to the loss of importance of the pandemic to the rural general population.

The overall seroprevalence in round one was 73.9% among which the highest was in the Gorakhpur site. The Gorakhpur site being in a busy international transit point with Nepal and the rapid transmission in the whole state of Uttar Pradesh might be the reason for high seroprevalence. The prevalence was slightly more in an urban area, among the older age (>18 years) group, and among those vaccinated with the SARS-CoV-2 vaccine but was not different between males and females. Our findings were consistent with the fourth nationwide serosurvey where the overall seroprevalence was 67%; and it was not different in urban and rural areas, between male and female but was higher among vaccinated participants [8]. The seroprevalence increased to 90.7% in round two. This time the prevalence in urban areas was closer to the rural area but was slightly higher among the older age group, vaccinated participants. A study done in the similar time period of round two in Delhi reported 89.5% overall

**Table 5. Distribution of participants by change in SARS-CoV-2 seropositivity status by round and area.**

| Baseline | Rural | | | | | | Urban | | | | | | Tribal | | | | | |
|---|---|---|---|---|---|---|---|---|---|---|---|---|---|---|---|---|---|---|
| | Round Two* | | | Round Three† | | | Round Two‡ | | | Round Three | | | Round Two§ | | | Round Three¶ | | |
| | n | Pos n (%) | Neg n (%) | n | Pos n (%) | Neg n (%) | n | Pos n (%) | Neg n (%) | n | Pos n (%) | Neg n (%) | n | Pos n (%) | Neg n (%) | n | Pos n (%) | Neg n (%) |
| Sero +ve | 2262 | 2222 (98.2) | 40 (1.7) | 2095 | 2071 (98.8) | 24 (1.2) | 2197 | 2157 (98.2) | 40 (1.8) | 1715 | 1672 (97.5) | 43 (2.5) | 317 | 308 (97.2) | 9 (2.8) | 293 | 291 (98.9) | 2 (0.6) |
| Sero -ve | 1028 | 756 (73.5) | 272 (26.5) | 219 | 100 (45.6) | 119 (54.3) | 512 | 310 (60.5) | 201 (39.3) | 166 | 43 (25.7) | 123 (73.6) | 169 | 133 (78.2) | 36 (21.2) | 26 | 9 (33.3) | 17 (62.9) |

* 5 borderline in first round got positive in round 2.

† 4 borderline in round 2 got positive in round 3

‡ Out of 6 borderline in round 1, 5 were positive and 1 were negative in round 2

§ Out of the 2 borderline in first round, 1 got positive and 1 got negative in round 2

¶ 1 borderline of round 2 got positive in round 3

seroprevalence among the general population with higher prevalence among the older age group, but little difference between male, female, urban, and rural areas [9]. Another study was done among children and adolescents in Delhi in October 2021, which found 81.8% prevalence among the age group <18 years but no difference between urban and rural areas, male and female participants [10]. In the third round, almost in every category, the seroprevalence was high. However, among the vaccinated individuals still seroprevalence was higher. This indicates that at the time of third wave of SARS-CoV-2 infection in India, almost all individuals had evidence of past infection regardless of age, sex, or area of residence.

We also explored the persistence of the immunity up to 6 months after a baseline seropositive result. Our study found that nearly the whole study cohort remained seropositive at the third month and six months in urban, rural, and tribal areas. This indicates the persistence of humoral immunity against SARS-CoV-2 infection for up to six months. A similar seroepidemiological cohort study done in Spain among already diagnosed cases also found that 99% of the participants had persistent antibodies at six months [11].

This was a nationwide multicentric seroepidemiological study involving participants from urban, rural, and tribal areas. The study was done in a close cohort across all the sites. Even after six months of follow-up during the challenging pandemic phase, there was a considerable number of participants remained in the cohort, and at the round, three more than 50% of participants were in the study.

Though we could assess total antibody against SRBD the neutralizing antibody couldn't be assessed. Moreover, our follow-up was limited up to six months in this current report. We couldn't assess the adaptive immunity also to have a comprehensive understanding of the total immunity over time. We could not do RTPCR among those who were found negative in previous rounds therefore the follow-up positive antibody could also be due to reinfection or re-exposure.

## Conclusion

This study explored the SARS-CoV-2 antibody prevalence in different sites, groups, and the antibody kinetics up to six months of baseline assessment. The antibody prevalence was high and increased over time. In most of the cases, the seroprevalence was not different based on the area, age group, sex, etc. However the vaccinated individual had a higher antibody prevalence. On the other hand, nearly all the participants had antibody positivity for up to six months. There is a further need for study for longer follow-up.

## Supporting information

**S1 File. S1 to S6 Tables on the supporting information of study result.**
(DOCX)

## Acknowledgments

We thank the WHO Country Office, India team particularly Mohammad Ahmad (National Professional Officer, WHO) and Anisur Rahman (Health Emergencies and Research Officer, WHO) for continuous support. We are immensely thankful to Meenu Sangral, Research Officer; Shreya Jha, Senior Research Consultant; Priyanka Kardam, Research Officer; Kapil Yadav, Professor; Bratati Pal, Research Officer; Tanushree Roy, Research officer for their support. Special thanks to the participants who allowed us to investigate the extent of infection, as determined by seropositivity in the general population, in which COVID-19 virus infection has been reported.

## Author Contributions

**Conceptualization:** Puneet Misra, Arvind Kumar Singh, Guruprasad R. Medigeshi, Hari Shanker Joshi, Mohammad Ahmad, Pradeep Kumar Chaturvedi, Partha Haldar, Mohan Bairwa, Randeep Guleria, Sanjay Kumar Rai, Shashi Kant, Subrata Baidya, Surekha Kishore, Vivek Hada.

**Data curation:** Arvind Kumar Singh, Baijayantimala Mishra, Guruprasad R. Medigeshi, Hari Shanker Joshi, Pradeep Kumar Chaturvedi, Palanivel Chinnakali, Pradeep Kharya, Rahul Dhodapkar, Ramashankar Rath, Randeep Guleria, Subrata Baidya, Suneeta Meena, Suprakash Mandal, Tapan Majumder.

**Formal analysis:** Puneet Misra, Baijayantimala Mishra, Suprakash Mandal.

**Funding acquisition:** Puneet Misra, Mohammad Ahmad.

**Investigation:** Arvind Kumar Singh, Baijayantimala Mishra, Bijayini Behera, Binod Kumar Patro, Guruprasad R. Medigeshi, Mohammad Ahmad, Pradeep Kumar Chaturvedi, Palanivel Chinnakali, Partha Haldar, Mohan Bairwa, Pradeep Kharya, Rahul Dhodapkar, Ramashankar Rath, Sitanshu Sekhar Kar, Sonali Sarkar, Subrata Baidya, Suneeta Meena, Tapan Majumder, Vivek Hada.

**Methodology:** Puneet Misra, Baijayantimala Mishra, Hari Shanker Joshi, Mohammad Ahmad, Pradeep Kumar Chaturvedi, Partha Haldar, Mohan Bairwa, Pradeep Kharya, Ramashankar Rath, Randeep Guleria, Sanjay Kumar Rai, Sitanshu Sekhar Kar, Shashi Kant, Sonali Sarkar, Subrata Baidya, Suneeta Meena, Suprakash Mandal, Surekha Kishore, Tapan Majumder, Vivek Hada.

**Project administration:** Puneet Misra, Arvind Kumar Singh, Mohammad Ahmad, Ramashankar Rath, Randeep Guleria, Sanjay Kumar Rai, Shashi Kant, Subrata Baidya, Suprakash Mandal, Surekha Kishore.

**Resources:** Puneet Misra, Mohammad Ahmad, Randeep Guleria, Surekha Kishore.

**Software:** Puneet Misra, Suprakash Mandal.

**Supervision:** Puneet Misra, Arvind Kumar Singh, Bijayini Behera, Binod Kumar Patro, Guruprasad R. Medigeshi, Hari Shanker Joshi, Mohammad Ahmad, Pradeep Kumar Chaturvedi, Palanivel Chinnakali, Ramashankar Rath, Randeep Guleria, Sanjay Kumar Rai, Shashi Kant, Subrata Baidya, Suprakash Mandal, Surekha Kishore.

**Validation:** Puneet Misra, Baijayantimala Mishra, Guruprasad R. Medigeshi, Hari Shanker Joshi, Mohammad Ahmad, Pradeep Kumar Chaturvedi, Palanivel Chinnakali, Rahul Dhodapkar, Sanjay Kumar Rai, Shashi Kant, Suneeta Meena, Suprakash Mandal, Surekha Kishore, Tapan Majumder, Vivek Hada.

**Visualization:** Puneet Misra, Randeep Guleria, Sanjay Kumar Rai, Shashi Kant, Suprakash Mandal.

**Writing – original draft:** Suprakash Mandal.

**Writing – review & editing:** Puneet Misra, Arvind Kumar Singh, Bijayini Behera, Binod Kumar Patro, Hari Shanker Joshi, Mohammad Ahmad, Pradeep Kumar Chaturvedi, Palanivel Chinnakali, Partha Haldar, Mohan Bairwa, Pradeep Kharya, Rahul Dhodapkar, Ramashankar Rath, Randeep Guleria, Sanjay Kumar Rai, Sitanshu Sekhar Kar, Shashi Kant, Sonali Sarkar, Subrata Baidya, Suneeta Meena, Suprakash Mandal, Surekha Kishore, Tapan Majumder, Vivek Hada.

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
