## [Decision Letter · Decision Letter 0]

23 Oct 2023

PONE-D-23-16775Anti SARS-CoV-2 Antibody Kinetics up to 6 months of follow-up: Result from a Nation-wide Population-based, Age Stratified Sero-Epidemiological Prospective Cohort Study in IndiaPLOS ONE

Dear Dr. Misra,

Thank you for submitting your manuscript to PLOS ONE. After careful consideration, we feel that it has merit but does not fully meet PLOS ONE’s publication criteria as it currently stands. Therefore, we invite you to submit a revised version of the manuscript that addresses the points raised during the review process.

Your manuscript was reviewed by two experts in the field. Both identified some problems in your submission. Please review the attached comments and provide point-by-point responses.

We look forward to receiving your revised manuscript.

Kind regards,

Yury E Khudyakov, PhD

Academic Editor

PLOS ONE

Journal Requirements:

"This work was supported by a research grant (Ref No: 2020/1085497, Purchase Order: 202630166) from the WHO Country Office, New Delhi 110016, India."

3. Please expand the acronym "WHO" (as indicated in your financial disclosure) so that it states the name of your funders in full.

6. We notice that your supplementary tables are included in the manuscript file. Please remove them and upload them with the file type 'Supporting Information'. Please ensure that each Supporting Information file has a legend listed in the manuscript after the references list.

Reviewers' comments:

Reviewer's Responses to Questions

**Comments to the Author**

1. Is the manuscript technically sound, and do the data support the conclusions?

Reviewer #1: Yes

Reviewer #2: Yes

2. Has the statistical analysis been performed appropriately and rigorously? 

Reviewer #1: Yes

Reviewer #2: Yes

3. Have the authors made all data underlying the findings in their manuscript fully available?

Reviewer #1: Yes

Reviewer #2: Yes

4. Is the manuscript presented in an intelligible fashion and written in standard English?

Reviewer #1: Yes

Reviewer #2: Yes

5. Review Comments to the Author

Reviewer #1: It is a well conducted multi centric study to know the change in antibody prevalence.

The aim of the study was: Therefore, we aimed to assess the change in antibody prevalence and its kinetics in a closed cohort over six months across the country in different sub-population in India.

The authors have concluded that: There is a further need for study for longer follow-up and finding actual protection from subsequent infection.

A clarification is required that did the authors assess any protection from subsequent infection in their study? If it is not assessed then this conclusion needs to be deleted.

Reviewer #2: 1) It would be appropriate if the results may be presented with respect to the IgM and IgG antibody prevalence for each site.

2) Authors to mention in the manuscript whether the 3 rounds of sample collection are from the same subjects or different people in the text, as the prevalence rate difference at various sites may be due to different times of sample collection, then the site does not have any relevance.

3) The data may be represented in more precise manner and the repeated data presentation in different forms without any new information may be avoided.

4) The reason for overlapping of sample collection during 2nd and 3rd rounds may be discussed in manuscript.

5) The manuscript may be improved by critically thinking the importance and context in which the findings stand for scientific community.

6) the manuscript may be written with description of the context specific results along with some insight on some population indicators like indigenous, migrated, work profile etc.

6. PLOS authors have the option to publish the peer review history of their article (what does this mean?). If published, this will include your full peer review and any attached files.

Reviewer #1: No

Reviewer #2: **Yes: **Dr. Suresh Yadav

---

## [Author Response · Author response to Decision Letter 0]

15 Nov 2023

The editorial and reviewer's comments were appropriate and was immensely helpful to improve the quality of the manuscript. I express my sincere thanks to you.

---

## [Editor Report · Decision Letter 1]

24 Nov 2023

Anti SARS-CoV-2 Antibody Kinetics up to 6 months of follow-up: Result from a Nation-wide Population-based, Age Stratified Sero-Epidemiological Prospective Cohort Study in India

PONE-D-23-16775R1

Dear Dr. Misra,

We’re pleased to inform you that your manuscript has been judged scientifically suitable for publication and will be formally accepted for publication once it meets all outstanding technical requirements.

Kind regards,

Yury E Khudyakov, PhD

Academic Editor

PLOS ONE
---

## [Editor Report · Acceptance letter]

30 Nov 2023

PONE-D-23-16775R1 

Anti-SARS-CoV-2 antibody kinetics up to 6 months of follow-up: Result from a nation-wide population-based, age stratified sero-epidemiological prospective cohort study in India 

Dear Dr. Misra:

I'm pleased to inform you that your manuscript has been deemed suitable for publication in PLOS ONE. Congratulations! Your manuscript is now with our production department. 

Kind regards, 

on behalf of

Dr. Yury E Khudyakov 

Academic Editor

PLOS ONE